# Robust olfactory responses in the absence of odorant binding proteins

**Shuke Xiao, Jennifer S Sun, John R Carlson\***

Department of Molecular, Cellular and Developmental Biology, Yale University, New Haven, United States

**Abstract** Odorant binding proteins (Obps) are expressed at extremely high levels in the antennae of insects, and have long been believed essential for carrying hydrophobic odorants to odor receptors. Previously we found that when one functional type of olfactory sensillum in *Drosophila* was depleted of its sole abundant Obp, it retained a robust olfactory response (Larter et al., 2016). Here we have deleted all the *Obp* genes that are abundantly expressed in the antennal basiconic sensilla. All of six tested sensillum types responded robustly to odors of widely diverse chemical or temporal structure. One mutant gave a greater physiological and behavioral response to an odorant that affects oviposition. Our results support a model in which many sensilla can respond to odorants in the absence of Obps, and many Obps are not essential for olfactory response, but that some Obps can modulate olfactory physiology and the behavior that it drives.
DOI: https://doi.org/10.7554/eLife.51040.001

## Introduction

Animals depend critically on olfactory systems to interpret the quality, quantity, and temporal structure of chemical cues in their environments. The molecules, neurons, and circuits underlying olfaction are subjects of intense investigation in many animals, including the fruit fly *Drosophila* (*Leal, 2013*; *Su et al., 2009*; *Wilson, 2013*).

The *Drosophila* antenna detects an enormous variety of odorants via olfactory sensilla on the antennal surface (*Figure 1A–C*). One morphological class of sensilla, the basiconic sensilla, is sensitive to many fruit odors (*de Bruyne et al., 2001*; *Dweck et al., 2018*; *Hallem and Carlson, 2006*). There are 10 functionally distinct types of antennal basiconic (ab) sensilla, termed ab1-ab10 (*Couto et al., 2005*; *de Bruyne et al., 2001*; *Grabe et al., 2016*). Each of these sensilla contains pores through which odorants can pass, and each contains the dendrites of two or four olfactory receptor neurons (ORNs) immersed in an aqueous lymph (*Figure 1D*) (*Shanbhag et al., 1999*; *Shanbhag et al., 2000*). The ORNs within a sensillum are distinguishable based on their spike amplitudes and response spectra, and are designated by letters, for example the ab2 sensillum contains ORNs ab2A and ab2B.

What is the molecular foundation on which olfactory detection, discrimination, and behavior ultimately rest? How do the molecular components of the olfactory system encode the identity, intensity, and temporal parameters of an olfactory stimulus? Two major classes of molecules are believed essential to this process, odor receptors and odorant binding proteins (Obps) (*Leal, 2013*). The in vivo functions of many odorant receptors have been well established (*Ai et al., 2010*; *Benton et al., 2009*; *Ha and Smith, 2006*; *Hallem and Carlson, 2006*; *Hallem et al., 2004*; *van der Goes van Naters and Carlson, 2007*; *Yao et al., 2005*), but their structures are only now beginning to be solved, with the recent determination of a cryo-EM structure for an odor receptor co-receptor, Orco (*Butterwick et al., 2018*). Reciprocally, the structures of many Obps have been solved at high resolution (*Brito et al., 2016*; *Kruse et al., 2003*; *Laughlin et al., 2008*; *Leite et al., 2009*; *Sandler et al., 2000*), but their functions in vivo are much less clear (*Bentzur et al., 2018*; *Gomez-*

**\*For correspondence:**
john.carlson@yale.edu

**Competing interests:** The authors declare that no competing interests exist.

*Diaz et al., 2013*; *Kim et al., 1998*; *Larter et al., 2016*; *Sun et al., 2018b*; *Xu et al., 2005*; *Zhang et al., 2017*).

Obps are intriguing in several respects. They are extraordinarily abundant, with *Obp* mRNAs being the most abundant in the *Drosophila* antenna (*Menuz et al., 2014*; *Vogt et al., 1989*). There are a great number of them, for example there are 52 *Obp* genes in *Drosophila melanogaster* (*Vieira and Rozas, 2011*). Obps are widely divergent in sequence (*Hekmat-Scafe et al., 2002*), suggesting they may be divergent in function. Despite their variable sequence, their overall structure is conserved. Obps are small proteins on the order of 14 kDa, with six conserved cysteines that form three disulfide bridges linking six α-helices (*Graham and Davies, 2002*; *Leal et al., 1999*).

In vitro, many Obps bind odorants. The binding characteristics of Obps from widely diverse insects have been described in detail (*Brito et al., 2016*; *Leal, 2013*; *Leal et al., 2005*; *Ziegelberger, 1995*). Many Obps are found in the aqueous lymph of the olfactory sensilla (*Klein, 1987*; *Shanbhag et al., 2005*; *Vogt and Riddiford, 1981*). Their odor-binding properties, location, and abundance have led to a model in which they provide an efficient solution to a difficult problem: the transport of hydrophobic odorants through the aqueous sensillar lymph (*Kaissling, 2009*; *Klein et al., 1987*; *Leal, 2013*; *van den Berg and Ziegelberger, 1991*; *Vogt, 1991*; *Zhou et al., 2004*). This model has been widely accepted for more than 30 years.

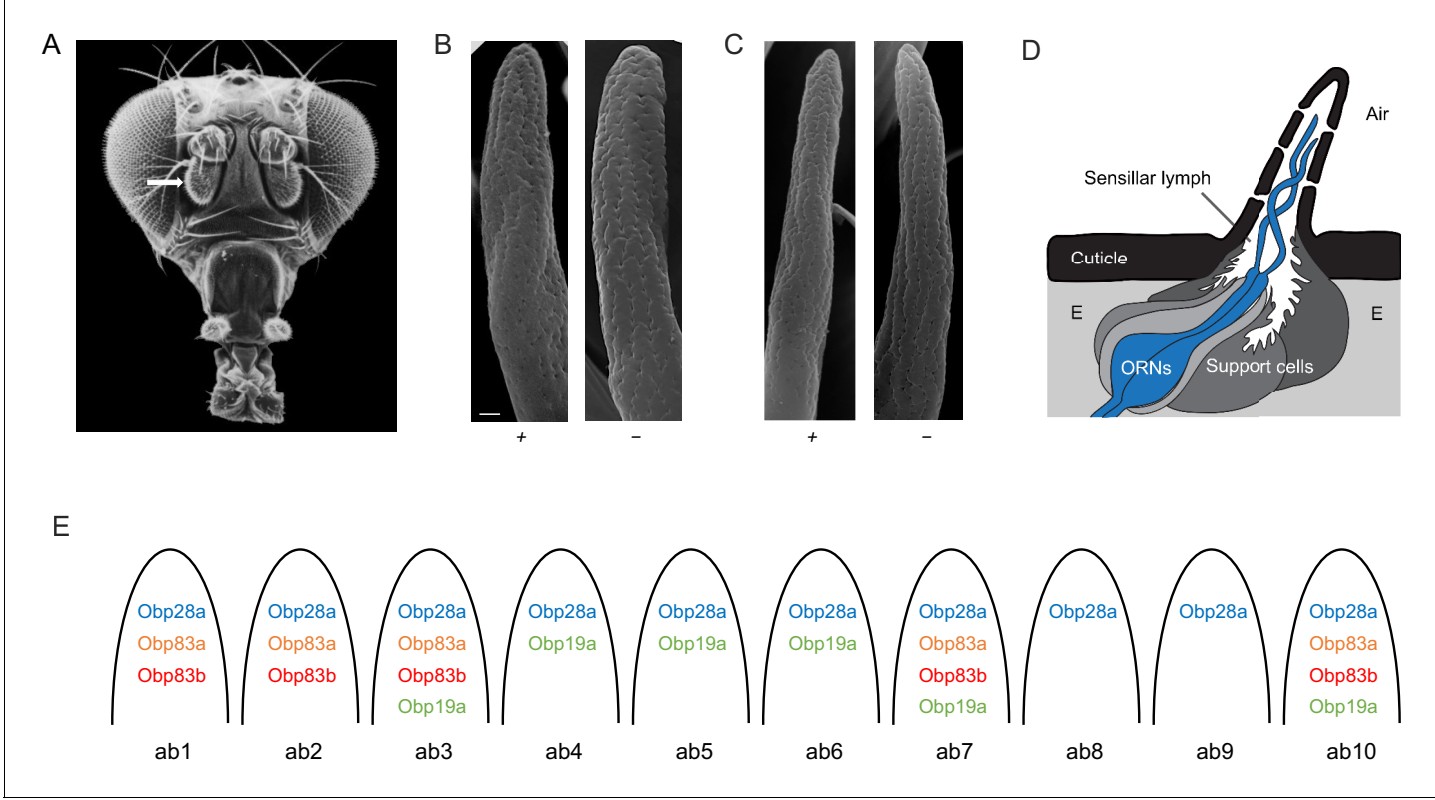

**Figure 1.** Organization of the *Drosophila* antenna and morphology of mutant sensilla. (A) Fly head. Arrow indicates antenna. Adapted from *Menuz et al. (2014)*, with thanks to Dr Rudi Turner. (B) Large basiconic sensilla of control (left) and of an *Obp19a⁻*; *Obp28a⁻*; *Obp83a⁻,b⁻* mutant (right). Scale bar is 0.5 µm and also applies to (C). (C) Thin basiconic sensilla of control (left) and an *Obp19a⁻*; *Obp28a⁻*; *Obp83a⁻,b⁻* mutant (right). (D) Olfactory sensillum. Adapted from *Larter et al. (2016)* and *Shanbhag et al. (1999)*. E = Epidermal cells. (E) Obp-to-sensillum map of the 10 functional types of basiconic sensilla.
DOI: https://doi.org/10.7554/eLife.51040.002
The following figure supplement is available for figure 1:

**Figure supplement 1.** Sensilla of mutant and control animals appear similar.
DOI: https://doi.org/10.7554/eLife.51040.003

A major barrier to studying the in vivo function of Obps has been that their distribution in the antenna had not been determined comprehensively at high resolution, in *Drosophila* or in any other insect. We recently overcame this barrier by constructing an Obp-to-sensillum map of the abundant antennal Obps, that is 10 Obps whose RNAs were expressed at much higher levels than all others in the antenna (*Figure 1E*) (*Larter et al., 2016*). The map was established via double-label experiments using *Obp* in situ hybridization probes and well-defined *Or-GAL4* drivers that label particular sensillum types. The map provides a basis for constructing defined manipulations of Obps.

The map revealed that one type of sensillum, ab8, contained a single abundant Obp, Obp28a (*Larter et al., 2016*). We deleted the *Obp28a* gene and were startled to find that the ab8 sensillum retained robust olfactory responses. We tested a wide variety of odorants and found that responses to odor pulses were either normal or, in some cases, of somewhat greater magnitude. These data indicated that Obp28a is not essential in ab8 for robust olfactory response to a wide variety of compounds, and that the ab8 sensillum does not require its sole abundant Obp in order to respond robustly.

These results were surprising and provocative, but they left open three critical possibilities. First, Obp28a could be an exceptional Obp. Obp28a is the only Obp that is expressed in all basiconic sensilla, suggesting that its function might not be representative of other Obps. Second, ab8 might be a singular sensillum. Third, Obps could be essential for transport of odorants of very high hydrophobicity; none of the odorants known to elicit responses from ab8 were highly hydrophobic.

In the present study we address all three of these possibilities. First, we have now deleted all of the abundant *Obp* genes that are expressed in basiconic sensilla, allowing us to test the roles of all of them. Second, we recombined the deletions to construct double, triple, and quadruple mutants, allowing testing of six functional types of sensilla in the absence of all of their abundant Obps. Third, we have now tested a wide variety of olfactory stimuli, including some of higher hydrophobicity than any tested in the previous genetic analysis. We have also tested stimuli that vary in their temporal dynamics. We find that in all mutants and in all sensilla, responses to all stimuli are robust as measured electrophysiologically. Responses are of normal magnitude in most cases; in no case is the response reduced. One odor elicits a greater electrophysiological response in a mutant sensillum, and correspondingly drives a greater behavioral response in the mutant. Taken together, these results provide extensive support for a model in which many sensilla can respond to odorants in the absence of their Obps. The evidence also supports the conclusion that many Obps are not essential for an olfactory response, but that some Obps can modulate both olfactory physiology and behavior.

## Results

### Robust responses to odor pulses in six sensillum types lacking abundant Obps

Different basiconic sensilla contain different subsets of four abundant Obps: Obp19a, Obp28a, Obp83a, and Obp83b (*Figure 1E*) (*Larter et al., 2016*). We have constructed deletions of all four of these genes via CRISPR-Cas9 genome editing. All of the deletions were outcrossed for five generations to a control genotype, Canton-S $w^{1118}$, in order to minimize genetic background effects. We then crossed the deletion mutations to each other so as to create mutants that are lacking the following combinations of Obps: Obp19a and Obp28a, which are the two abundant Obps expressed in ab4, ab5 and ab6; Obp28a, Obp83a and Obp83b, the three Obps expressed in ab1 and ab2; and Obp19a, Obp28a, Obp83a and Obp83b, the four Obps expressed in ab3. The absence of the deleted DNA was verified by PCR analysis at least three times: after construction of the original deletions, after backcrossing to Canton-S $w^{1118}$, and after recombining the mutations together.

We carried out a morphological examination of the mutant sensilla by scanning electron microscopy, which was not done in the previous study (*Larter et al., 2016*). On the order of 1000 basiconic sensilla were examined from the antennae of ~20 mutant flies lacking all four Obps. No gross defects in size, morphology, or position were detected (*Figure 1B,C* and *Figure 1—figure supplement 1*).

We then tested the physiological effects of eliminating all abundant Obps from each of these six sensillum types, using single-unit electrophysiology. We initially tested each mutant sensillum with one odorant that elicits relatively strong responses at relatively low concentrations from the A

neuron, and another such odorant for the B neuron. Each odorant was presented as a 0.5 s pulse, at two concentrations. In each case we compared the response of either a double, triple, or quadruple mutant to that of the control.

The collection of odorants tested in this initial analysis represents a variety of chemical classes, including alcohols, ketones, aldehydes, esters, terpenes and aromatics. Their hydrophobicity varies over more than five orders of magnitude, ranging from 2,3-butanedione (logP = −1.33, where P is the partition coefficient between octanol and water) to geranyl acetate, the most hydrophobic (logP = 4.10)(*Figure 2—figure supplement 1*). The tested ORNs range from broadly tuned, for example ab3A, which responds to many odorants (*Hallem and Carlson, 2006*; *Münch and Galizia, 2016*), to narrowly tuned, for example ab4B, which responds to geosmin (*Stensmyr et al., 2012*). The control responses to the odor stimuli in this experiment also extended over a broad range, from modest (~20 spikes/s) to very strong (~220 spikes/s).

ab1 sensilla lacking their three abundant Obps showed strong responses to ethyl acetate, which activates the ab1A neuron (*Figure 2A*). Response magnitudes of the mutant were indistinguishable from those of the genetic background control, at each of two concentrations. Responses were also normal when tested with two concentrations of 2,3-butanedione, which activates the ab1B neuron (*Figure 2A*).

ab2 sensilla likewise showed normal responses of both A and B neurons when these three Obps were removed genetically (*Figure 2B*). In this case methyl acetate was used to activate the ab2A neuron, and ethyl 3-hydroxybutyrate activated the ab2B neuron.

ab3 sensilla were tested following removal of all four of their abundant Obps. Responses of both ab3A and ab3B neurons to the test odorants were normal, compared to the genetic background control (*Figure 2C*).

ab4, ab5, and ab6 were tested in *Obp19a; Obp28a* double mutants. Responses to A and B neurons were normal in each of the three sensillum types (*Figure 2D–F*). We note that geosmin and geranyl acetate, which elicited normal responses, are both highly hydrophobic (*Figure 2—figure supplement 1*). The lower concentration of geosmin evoked a response of ~25 spikes/s and the higher concentration of geranyl acetate elicited a response of ~220 spikes/s, from ab4B and ab5A respectively.

To determine whether we could identify an olfactory response that was affected by the loss of Obps, we screened an additional 73 odorant-ORN combinations. These odorants included two that are more hydrophobic than any in the original set: geranyl acetone (logP = 4.13) and citronellyl acetate (logP = 4.28). All showed normal responses to a 0.5 s pulse of odorant in either the initial screen at a low n value (*Figure 2—figure supplement 2A–D*) or, in the case of 3 of the 73 odorants, in a rescreen at a higher n value (*Figure 2—figure supplement 2E*; the rescreen also included five control odorants). We note finally that the spontaneous activity level of all 12 neurons was normal in the absence of abundant Obps (p>0.05 in all cases, Mann-Whitney U test, n = 10).

## Robust responses to intermittent and prolonged odor stimuli in sensilla lacking Obps

The preceding results were obtained using a single 0.5 s pulse of odorant. Responses were measured by counting the total number of spikes produced by the pulse during 0.5 s. We wondered whether any of these Obps might play an essential role in response to a different kind of olfactory stimulus. Alternatively, perhaps Obps play a role that would be discernable via an alternative means of analysis. For example, if Obps act in odorant transport or removal, which have both been proposed (*Kaissling, 2009*; *Leal, 2013*; *Pelosi et al., 2018*), the dynamics of odor response might be affected by loss of Obps.

In natural environments, flies often encounter olfactory stimuli in the form of odor plumes (*Murlis et al., 1992*), in which the stimulus reaches the antenna intermittently. We provided an intermittent pattern of stimulation by delivering a series of 0.5 s pulses at a frequency of 1 Hz (*Figure 3A*). We then recorded the spikes that were elicited and plotted a peri-stimulus time histogram (PSTH) of the spikes in 25 ms intervals. For this analysis we used ab3 sensilla, which contain four Obps, and tested the mutant that lacks all four of them. We used the hydrophobic odorants ethyl hexanoate and 2-heptanone (logP = 2.38 and logP = 1.97, respectively) to activate the ab3A and ab3B neurons.

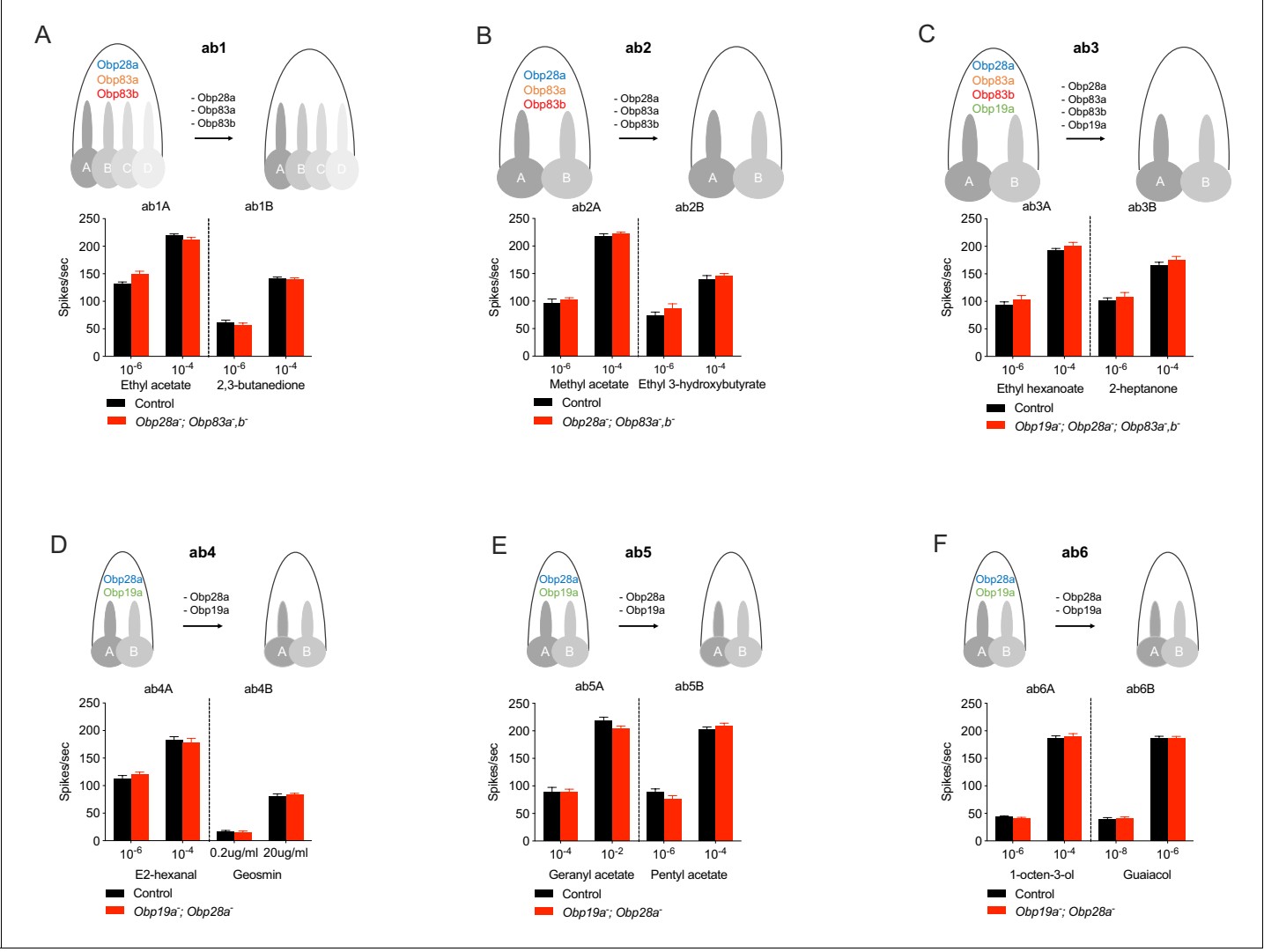

**Figure 2.** Responses of individual ORNs in each type of sensillum after removal of abundant Obps. (**A–F**) ab1-ab6. Responses are to 0.5 s pulses of strong ligands. (SEM; n = 6). No significant differences were found by the Mann-Whitney U test.

DOI: https://doi.org/10.7554/eLife.51040.004

The following source data and figure supplements are available for figure 2:

**Source data 1.** Source data for spike numbers in *Figure 2*.
DOI: https://doi.org/10.7554/eLife.51040.009
**Figure supplement 1.** Hydrophobicity of odorants.
DOI: https://doi.org/10.7554/eLife.51040.005
**Figure supplement 1—source data 1.** Source data for logP values plotted in *Figure 2—figure supplement 1*.
DOI: https://doi.org/10.7554/eLife.51040.006
**Figure supplement 2.** Screen of additional odorants against ORNs in mutant and control basiconic sensilla.
DOI: https://doi.org/10.7554/eLife.51040.007
**Figure supplement 2—source data 1.** Source data for spike numbers in *Figure 2—figure supplement 2*.
DOI: https://doi.org/10.7554/eLife.51040.008

Consistent with results using a single pulse, ab3A responded robustly to each of the repeated odor pulses in the mutant (*Figure 3A,C*). Moreover, the dynamics of the response throughout the entire stimulus period appeared comparable to that of the control. We also tested ab3A with 2-pentanone, which in control flies elicits a strong response with different dynamics: the ab3A response to 2-pentanone shows faster adaptation than the response to ethyl hexanoate. The response of the

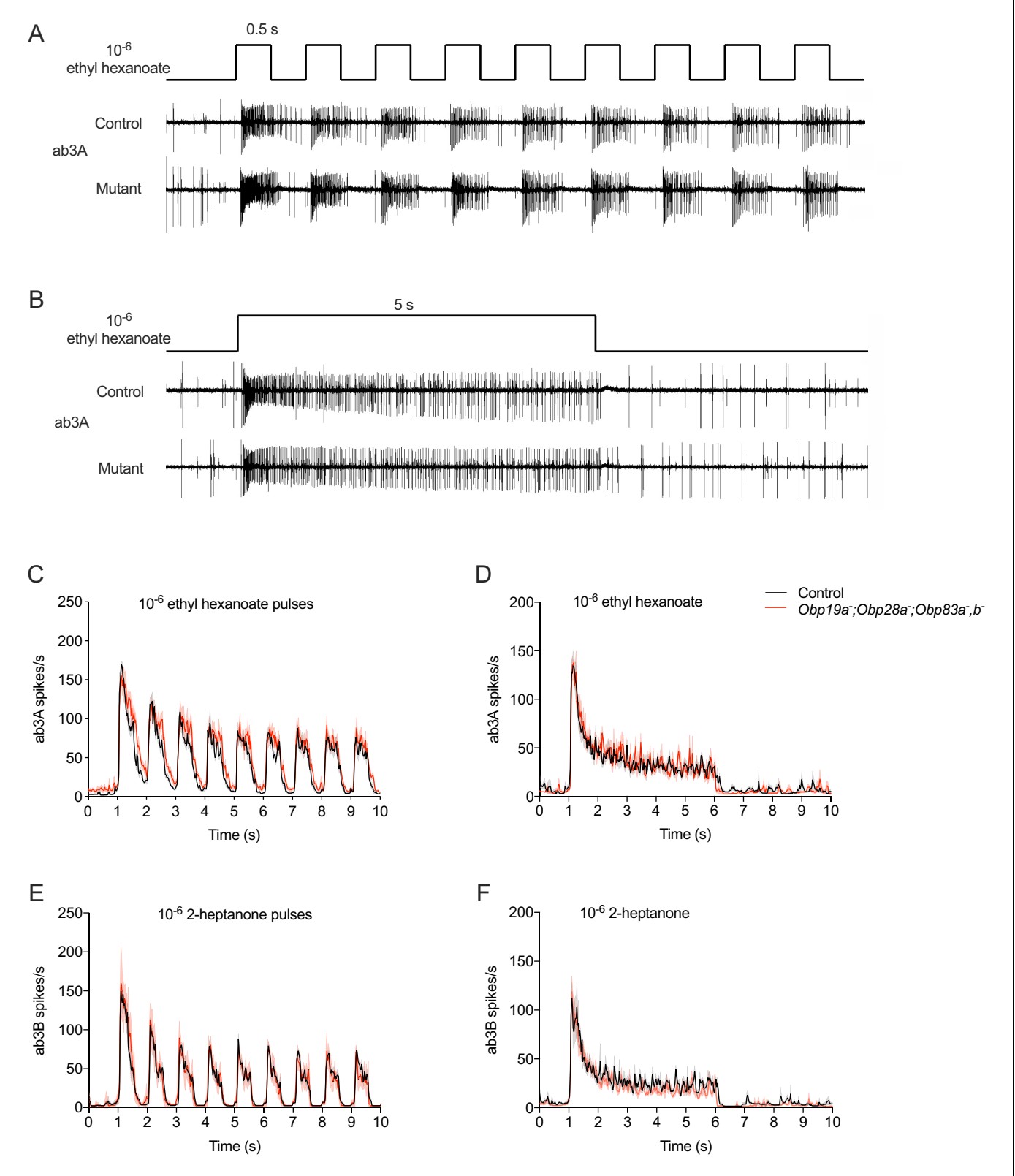

**Figure 3.** Responses to stimuli of different temporal dynamics in mutant ab3 sensilla. (**A**) Consecutive 500 ms pulses and representative responses to ethyl hexanoate, which activates ab3A, in control and $Obp19a^-$; $Obp28a^-$; $Obp83a^-,b^-$ flies. (**B**) Responses to a 5 s pulse. (**C**) Peri-stimulus time histogram (PSTH) of ab3A responses to consecutive 500 ms pulses of $10^{-6}$ ethyl hexanoate delivered at 1 Hz. (**D**) A PSTH of ab3B responses to a 5 s pulse of $10^{-6}$

*Figure 3 continued on next page*

*Figure 3 continued*

2-heptanone, n = 4. (E) PSTH of ab3B responses to consecutive 500 ms pulses of $10^{-6}$ 2-heptanone, n = 5. (F) PSTH of ab3A responses to a 5 s $10^{-6}$ ethyl hexanoate pulse, n = 5. The bin sizes for PSTHs is 25 ms. Shaded areas surrounding each curve indicate SEM.

DOI: https://doi.org/10.7554/eLife.51040.010

The following source data and figure supplements are available for figure 3:

**Source data 1.** Source data for PSTH numbers in *Figure 3*.

DOI: https://doi.org/10.7554/eLife.51040.013

**Figure supplement 1.** Responses in ab3 of *Obp19a⁻; Obp28a⁻; Obp83a⁻,b⁻* flies to 2-pentanone in two odor stimulus patterns.

DOI: https://doi.org/10.7554/eLife.51040.011

**Figure supplement 1—source data 1.** Source data for PSTH numbers in *Figure 3—figure supplement 1*.

DOI: https://doi.org/10.7554/eLife.51040.012

mutant to 2-pentanone was again comparable to the control (*Figure 3—figure supplement 1A*). The other neuron in the ab3 sensillum, ab3B, also showed robust responses to repeated pulses of an odorant, 2-heptanone. The response dynamics again appeared similar to the control (*Figure 3E*).

As another sensitive test of odor response, we challenged the mutant ab3 'empty sensillum' with another kind of stimulus dynamics found in nature: a prolonged pulse of odor. We delivered a 5 s pulse of ethyl hexanoate or 2-heptanone and measured the instantaneous spike rate (*Figure 3B,D, F*). The response of the mutant was very similar to that of control during the rising phase, during the peak period, and during the falling phase. We also found normal responses to a prolonged 2-pentanone stimulus (*Figure 3—figure supplement 1B*).

In summary, these ORNs are able to respond comparably to controls when tested with both intermittent and prolonged odor stimuli in the absence of their four abundant Obps.

## Increased physiological and behavioral responses in an ab4 mutant

The ab4A neuron and its receptor Or7a are broadly tuned and respond strongly to a wide variety of short chain aldehydes and alcohols (*Hallem and Carlson, 2006*; *Münch and Galizia, 2016*). This neuron and receptor have been shown to drive an oviposition preference behavior (*Lin et al., 2015*). ab4A has also been reported to respond to samples of several long chain compounds that are very hydrophobic, including linoleic acid (*Hallem and Carlson, 2006*), $(Z)-9$-tricosene (*Lin et al., 2015*) and bombykol (*Syed et al., 2006*). We decided to test the physiological and behavioral responses to a sample of linoleic acid, a compound that is more hydrophobic than any tested previously in this study but which is sufficiently soluble in agarose that it can be used conveniently in a well-established oviposition paradigm.

We found that the linoleic acid sample elicited a stronger response from the mutant ab4 sensillum than from a control sensillum (*Figure 4A*), reminiscent of the greater responses we observed previously with some odorants in the mutant ab8 sensillum (*Larter et al., 2016*). The ab4 mutant, *Obp19a; Obp28a,* lacks both of the abundant Obps normally present in ab4. The greater response is observed across a broad range of dilutions of the odor (*Figure 4A*).

We wondered whether the greater electrophysiological response in the Obp mutant would translate into a greater behavioral response. We used a two-choice oviposition preference test (*Figure 4B*) in which 10 mated females were allowed to lay eggs for 20–24 hr on a plate containing four quadrants, one containing agarose with the linoleic acid samples, and one containing agarose alone. At the end of the assay an oviposition preference index (PI) was calculated based on the number of eggs on each of these two quadrants.

Control flies showed a preference for laying eggs on low concentrations of the odor, as opposed to agarose without odor, a preference that gradually increased with increasing concentration, but which became aversive at the highest concentrations (*Figure 4C*). The Obp mutant flies also showed a preference at low odor concentrations, but with a lower threshold than the control flies: the mutant showed a response to lower concentrations than the control. Likewise, lower concentrations elicited aversion from the mutant than the control. The entire dose-response relationship is shifted in the mutant.

These results show that loss of Obps from ab4 can increase olfactory and oviposition responses. We note that we have referred to the olfactory stimulus as a sample of linoleic acid, rather than

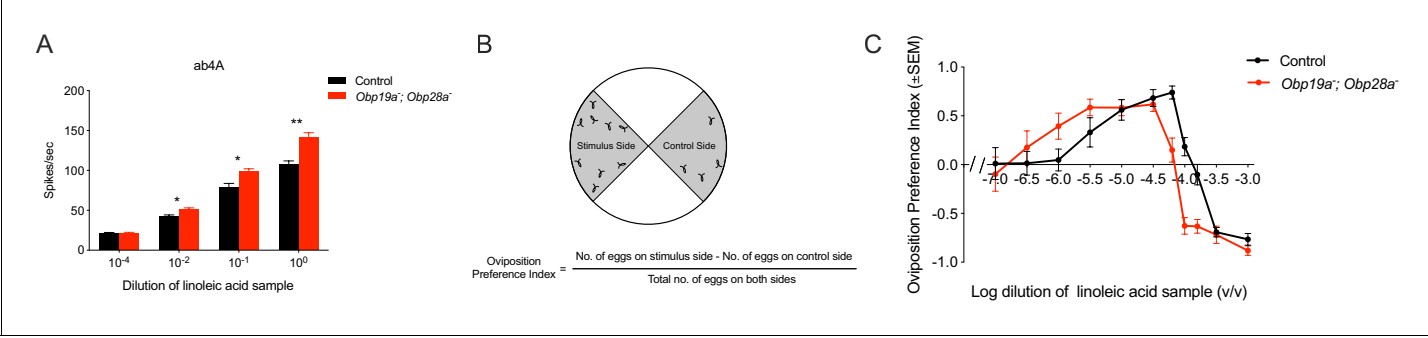

**Figure 4.** *Obp19a⁻; Obp28a⁻* flies have stronger responses to a sample of linoleic acid. (**A**) Physiological responses of ab4 sensilla to a sample of linoleic acid are stronger across several concentrations than control responses. * p 0.05, ** p 0.01, non-parametric t test, n = 5. (**B**) Two-choice oviposition preference paradigm. (**C**) Shift in oviposition preference for a linoleic acid sample in *Obp19a⁻; Obp28a⁻* mutants.

DOI: https://doi.org/10.7554/eLife.51040.014

The following source data is available for figure 4:

**Source data 1.** Source data for oviposition preference index numbers in *Figure 4*.
DOI: https://doi.org/10.7554/eLife.51040.015

linoleic acid per se. After our experiments were conducted, evidence became available that the physiological responses of ab4A to long chain hydrophobic compounds may arise from trace impurities in commercial preparations of these compounds (*Schorkopf et al., 2018*). Trace impurities are a potential issue in most olfactory research. The principal conclusion from our analysis, however, remains that loss of Obps from ab4 can affect function.

Taken together, the simplest interpretation of our results is that in the ab4 sensillum, Obps can modulate the olfactory response, and in their absence a greater physiological response of the ab4 sensillum drives a stronger behavioral response.

## Discussion

In this study we have taken advantage of detailed molecular and cellular maps of the *Drosophila* olfactory system to investigate the role of Obps in odor coding. Several conclusions emerge.

From the perspective of olfactory sensilla, all of the six tested types of basiconic sensilla responded robustly to all tested odorants despite removal of their abundant Obps. In no case was an olfactory response diminished by the absence of these Obps. These results were obtained using three different mutant genotypes: a double mutant, a triple mutant, and a quadruple mutant.

From the perspective of Obps, none was essential for the production of a robust olfactory response. In no case did a response decrease for lack of an individual Obp. These Obps are highly divergent, for example Obp83a shows only 21.3% identity to Obp28a and 15.6% sequence identity to Obp19a.

Our analysis included 37 odorants of diverse structures, and 85 odorant-ORN combinations. Four of the odorants are more hydrophobic (logP values ranging from 3.85 to 4.28) than the most hydrophobic odorant tested in our previous study, 1-octanol (logP = 3.07), arguing against the possibility that Obps might be essential to transport more hydrophobic odorants through the aqueous sensillar lymph. Responses to olfactory stimuli of diverse temporal structures were also robust.

In the case of the ab4 sensillum, loss of its Obps, that is Obp19a and Obp28a, caused an increase in physiological response to an odor stimulus. This result is consistent with the gain in response observed following loss of Obp28a in the ab8 sensillum (*Larter et al., 2016*). Here we have extended the analysis to the behavioral level, and found that the increase in physiological response is accompanied by an increase in oviposition response. The simplest interpretation of these results is that the Obp modulates both the physiological response and the behavioral response.

We note some formal possibilities. First, all of the data in this study were collected after the mutations had been backcrossed to the control stock for five generations and combined with each other. It is conceivable that the loss of each Obp initially led to a reduction in olfactory response, but that this phenotype was quickly suppressed during the several generations of genetic crosses that

ensued. In principle, suppression of nearly any phenotype can occur in *Drosophila* genetics; in the present case, if such suppression occurred it must have been complete, as opposed to partial, and must have occurred in all sensilla, in all mutants tested, during a limited number of generations, a constellation of requirements that collectively seem unlikely.

Second, it is formally possible that the Obps we have eliminated are redundant in function with other Obps that are expressed at dramatically lower levels. The Obp-to-sensillum map was based on in situ hybridization of *Obps* detected in the antenna by RNA-seq (*Larter et al., 2016*; *Menuz et al., 2014*). There are a number of antennal *Obp* RNAs that are expressed at much lower levels–two or three orders of magnitude lower in nearly all cases–than those of *Obp19a*, *Obp28a*, *Obp83a* and *Obp83b*. These *Obps* either did not yield a signal by in situ hybridization or were untested. We cannot exclude the possibility that these rare *Obps* provide function in the absence of the abundant *Obps*. However, such function would need to be provided in all six mutant sensilla, which collectively lack all four abundant *Obps*; moreover the compensation would need to be complete, not partial.

Third, we acknowledge that our analysis is not exhaustive. We have tested a wide diversity of odorants, and 12 different ORNs, but for any particular odorant-ORN combination we have tested at most three concentrations. Although we have tested stimuli that elicit a wide range of ORN activity, it is possible that a full dose-response analysis of all odorants would identify a concentration of one at which an *Obp* mutant showed a reduced response. Also, we have not analyzed the response dynamics extensively; in particular it is possible that shorter or longer pulses of odorants might reveal a strong phenotype.

The interpretation that *Obp19a*, *Obp28a*, *Obp83a* and *Obp83b* are not essential for the transport of all odorants in all sensilla is consistent with the results of some other recent studies. First, Obp28a and its tsetse ortholog were found to play a role outside the olfactory system, in hematopoiesis and wound-healing (*Benoit et al., 2017*). Second, another abundant antennal Obp, Obp59a, was discovered to act in humidity detection (*Sun et al., 2018a*). These studies provide precedent for functions of abundant antennal Obps other than carrying odorants to odor receptors. Many other *Obp* genes of *Drosophila* or other insects are expressed outside the olfactory system (*Arya et al., 2010*; *Galindo and Smith, 2001*; *Jeong et al., 2013*; *Li et al., 2005*; *Matsuo et al., 2007*; *Shanbhag et al., 2001*), again consistent with diverse roles for this large gene family.

We note further that genomic analysis has revealed dramatic expansions of odorant receptor gene families in some insect species, including a variety of Hymenopterans (*Missbach et al., 2014*; *Robertson, 2019*). In many cases this expansion is not accompanied by a concomitant expansion of Obps (*Vieira and Rozas, 2011*). For example, the eusocial ants *Camponotus floridanus* and *Pogonomyrmex barbatus* contain 407 and 399 Ors, respectively (*Bonasio et al., 2010*; *Zhou et al., 2012*), but only 13 and 16 Obps (*McKenzie et al., 2014*). These major imbalances argue against some classic models of odorant-Obp-receptor interactions.

In our previous study of the ab8 sensillum, we found that removal of Obp28a, its sole abundant Obp, produced greater responses to pulses of some but not all odorants, which we suggested might reflect a role for Obps in a form of molecular gain control (*Larter et al., 2016*). Consistent with these previous findings, the present results with the mutant ab4 sensillum also revealed an increase in response to several concentrations of one stimulus, but not other odorants. Here we have extended these findings by showing that this modulation of physiological response is accompanied by a corresponding modulation of behavioral response.

In this study we have examined basiconic sensilla because we have a detailed Obp-to-sensillum map of them. A recent study of other kinds of antennal sensilla, trichoid sensilla and intermediate sensilla, also found that when *Obp83a* and *Obp83b* were deleted, responses to all tested odorants were robust (*Scheuermann and Smith, 2019*). In this study, however, the responses of three of ten tested neurons showed slow deactivation kinetics; one neuron was tested with two odorants and showed the deactivation phenotype with one odorant but not the other. This finding of a phenotype for some odorant-ORN combinations but not others is reminiscent of our finding of a phenotype with a linoleic acid sample in ab4, but not other odorant-ORN combinations.

The mechanisms underlying the phenotypes observed by *Scheuermann and Smith (2019)* and us are unclear. *Scheuermann and Smith (2019)* propose that after the affected odorants bind and activate receptors, the Obps then remove them. The phenotypes we have found with the linoleic acid sample are consistent with a model in which these Obps bind an odorant and reduce the concentration that reaches the ORN. However, what is perhaps most surprising from both of these studies is

that the genetic removal of such abundant proteins, whose synthesis is likely to incur a major metabolic cost, does not have more widespread effects on odorant response. The behavioral effect we observe with the linoleic sample is consistent with the physiological effect and may have a great deal of significance for the animal in a natural context. However, it seems clear that loss of Obps does not impair the ability of many sensilla to respond robustly to many odorants. The results raise the interesting possibility that some of these antennal Obps might play a role in processes other than olfaction, as they do in other organs.

In summary, this study greatly increases the body of evidence that Obps are not universally required for odorant transport in the *Drosophila* olfactory system. Our results extend our previous analysis from one to seven basiconic sensilla, from one to four Obps, and from 12 to 37 odorants, including several more hydrophobic than any tested before. The results confirm, however, that Obps can in some cases modulate the physiological response to certain olfactory stimuli, and we show that Obps can modulate a behavioral response as well.

# Materials and methods

## Key resources table

| Reagents type (species) or resource | Designation | Sourceor reference | Identifiers | Additional information |
|---|---|---|---|---|
| Gene (*Drosophila melanogaster*) | *Obp19a* | Flybase | FBgn0031109 | |
| Gene (*Drosophila melanogaster*) | *Obp28a* | Flybase | FBgn0011283 | |
| Gene (*Drosophila melanogaster*) | *Obp83a* | Flybase | FBgn0011281 | |
| Gene (*Drosophila melanogaster*) | *Obp83b* | Flybase | FBgn0010403 | |
| Strain (*Drosophila melanogaster*) | *Canton-S w[1118]* | *Koh et al., 2014* | NA | DOI: 10.1016/j.neuron.2014.07.012 |
| Strain (*Drosophila melanogaster*) | *Obp19a[-]* | This paper | NA | Materials and methods section |
| Strain (*Drosophila melanogaster*) | *Obp28a[-]* | *Larter et al., 2016* | NA | DOI: 10.7554/eLife.20242 |
| Strain (*Drosophila melanogaster*) | *Obp83a[-],b[-]* | This paper | NA | Materials and methods section |
| Recombinant DNA reagen | pCFD4 | Addgene | CAT # 49411 | pCFD4-U6:1_U6:3tandemgRNAs |
| Recombinant DNA reagen | pHD-DsRed-attP | Addgene | CAT # 51019 | |
| Chemical compound | Ethyl acetate | MilliporeSigma | CAT # 270989 | CAS # 141-78-6 |
| Chemical compound | 2,3-butanedione | MilliporeSigma | CAT # 11038 | CAS # 431-03-8 |
| Chemical compound | Methyl acetate | MilliporeSigma | CAT # 45999 | CAS # 79-20-9 |
| Chemical compound | Ethyl 3-hydroxybutyrate | MilliporeSigma | CAT # E30603 | CAS # 5405-41-4 |
| Chemical compound | Ethyl hexanoate | MilliporeSigma | CAT # 148962 | CAS # 123-66-0 |
| Chemical compound | 2-heptanone | MilliporeSigma | CAT # 537683 | CAS # 110-43-0 |
| Chemical compound | E2-hexanal | MilliporeSigma | CAT # 132659 | CAS # 6728-26-3 |

*Continued on next page*

*Continued*

| Reagents type (species) or resource | Designation | Sourceor reference | Identifiers | Additional information |
|---|---|---|---|---|
| Chemical compound | (±) Geosmin | MilliporeSigma | CAT # G5908 | CAS # 16423-19-1 |
| Chemical compound | Geranyl acetate | MilliporeSigma | CAT # 45896 | CAS # 105-87-3 |
| Chemical compound | Pentyl acetate | MilliporeSigma | CAT # 109584 | CAS # 628-63-7 |
| Chemical compound | 1-octen-3-ol | MilliporeSigma | CAT # O5284 | CAS # 3391-86-4 |
| Chemical compound | Guaiacol | MilliporeSigma | CAT # G5502 | CAS # 90-05-1 |
| Chemical compound | Linoleic acid | MilliporeSigma | CAT # L1376 | CAS # 60-33-3 |
| Chemical compound | Paraffin oil | MilliporeSigma | CAT # 18512 | CAS # 8012-95-1 |
| Chemical compound | Ethanol | MilliporeSigma | CAT # E7023 | CAS # 64-17-5 |
| Chemical compound | Acetone | Fisher Scientific | CAT # 9006–01 | CAS # 67-64-1 |
| Chemical compound | 2-pentanone | MilliporeSigma | CAT # W284203 | CAS # 107-87-9 |
| Chemical compound | 2-butanone | MilliporeSigma | CAT # 34861 | CAS # 78-93-3 |
| Chemical compound | 1-pentanol | MilliporeSigma | CAT # 398268 | CAS # 71-41-0 |
| Chemical compound | Ethyl propionate | MilliporeSigma | CAT # 112305 | CAS # 105-37-3 |
| Chemical compound | Ethyl lactate | MilliporeSigma | CAT # E34102 | CAS # 687-47-8 |
| Chemical compound | Methyl butyrate | MilliporeSigma | CAT # 246093 | CAS # 623-42-7 |
| Chemical compound | Ethyl butyrate | MilliporeSigma | CAT # 75563 | CAS # 105-54-4 |
| Chemical compound | 1-hexanol | MilliporeSigma | CAT # H13303 | CAS # 11-27-3 |
| Chemical compound | E3-hexenol | MilliporeSigma | CAT # 224715 | CAS # 928-97-2 |
| Chemical compound | Hexanal | MilliporeSigma | CAT # 115606 | CAS # 66-25-1 |
| Chemical compound | Hexanoic acid | MilliporeSigma | CAT # 153745 | CAS # 142-62-1 |
| Chemical compound | alpha-terpineol | MilliporeSigma | CAT # 30627 | CAS # 98-55-5 |
| Chemical compound | gamma-hexalactone | MilliporeSigma | CAT # 68554 | CAS # 695-06-7 |
| Chemical compound | 3-methyl-2-buten-1-ol | MilliporeSigma | CAT # 162353 | CAS # 556-82-1 |
| Chemical compound | 6-methyl-5-hepten-2-one | MilliporeSigma | CAT # M48805 | CAS # 110-93-0 |
| Chemical compound | E2-hexenol | MilliporeSigma | CAT # W256230 | CAS # 928-95-0 |
| Chemical compound | Z3-hexenol | MilliporeSigma | CAT # W2564307 | CAS # 928-96-1 |

*Continued on next page*

*Continued*

| Reagents type (species) or resource | Designation | Sourceor reference | Identifiers | Additional information |
|---|---|---|---|---|
| Chemical compound | Diethyl succinate | MilliporeSigma | CAT # 07429 | CAS # 123-25-1 |
| Chemical compound | Ethyl trans-2-butenoate | MilliporeSigma | CAT # W348630 | CAS # 623-70-1 |
| Chemical compound | Benzaldehyde | MilliporeSigma | CAT # 418099 | CAS # 100-52-7 |
| Chemical compound | E2-hexenyl acetate | MilliporeSigma | CAT # W256404 | CAS # 2497-18-9 |
| Chemical compound | Citronellyl acetate | MilliporeSigma | CAT # 43646 | CAS # 150-84-5 |
| Chemical compound | Geranyl acetone | MilliporeSigma | CAT # W354201 | CAS # 689-67-8 |
| Chemical compound | o-cresol | MilliporeSigma | CAT # 36922 | CAS # 95-48-7 |
| Chemical compound | Agarose | AmeicanBio | CAT # AB00972 | CAS9 9012-36-6 |
| Chemical compound | Sucrose | MilliporeSigma | CAT # S7903 | CAS # 57-50-1 |
| Software | Prism 7 | GraphPad Prism | RRID:SCR_002798 | |
| Software | AutoSpike32 | Syntech | | http://www.ocken fels-syntech.com |

## Generation of *Obp* mutants

The *Obp19a*⁻ mutant was generated using the guide RNA sequences (5'-GTCTGCGTCGCCATA TCCCT; 5'-GCTAGCTTTTATTGATCATC), using the same procedure that was described in *Larter et al. (2016)* to create *Obp28a*⁻. The deletion removed 91% of the *Obp19a* coding sequence. *Obp83a* and *Obp83b* are located within 1 kb of each other and were deleted together to create a mutation denoted *Obp83a*⁻,*b*⁻. The *Obp83a*⁻,*b*⁻ mutant was created using CRISPR-Cas9 in CAS-0001 flies ($y^2$ $cho^2$ $v^1$; attP40 nos-Cas9/CyO) (*Kondo and Ueda, 2013*), using the following guide RNA sequences: (5'-GCCCAGGAACCAAGGCGCGA; 5'-GCCATTTTCAGGATGCCCGG). The deletion removed 27% of the *Obp83a* coding region, including the first two exons and therefore the start codon; the deletion removed 85% of the *Obp83b* coding region. The *Obp28a* deletion was constructed previously (*Larter et al., 2016*). Plasmids carrying gRNAs and homology arms were injected by Bestgene, Inc (Chino Hills, CA). Deletions were verified using the following primers:

*Obp19a*⁻: 5'-ATGAAGTTCCATCTGCTGCTG; 5'-GATCCCTTTGTGCTTATGGAA.
*Obp28a*⁻: 5'-ATGCAGTCTACTCCAATCATTC; 5'-TTACAAGAGTCCATGTTTCTTG.
*Obp83a*⁻,*b*⁻: 5'-ACCCACTGATACTACTTTTGAT; 5'- CGCCCGTCTTCTCCAC.

All *Obp* deletion lines were outcrossed to Canton-S $w^{1118}$ for five generations before testing.

## Odor stimuli

Odorants were prepared as described previously (*Larter et al., 2016*). The sources and catalogue numbers of odorants are indicated in the *Key Resources Table*. A constant humidified air flow (~3000 ml/min) and stimulus pulses (~300 ml/min) were produced by a Stimulus Air Controller CS-55 V2 (Syntech). For the repeated odor pulses in *Figures 3A,C,E*, 500 ms pulses were repeated automatically, with 500 ms intervals between pulses. For long pulses in *Figure 3B,D,F*, the duration of the stimulus pulses was increased to 5 s.

## Electrophysiology

3–6 day old female flies were used for single-sensillum recordings, as described previously (*Dobritsa et al., 2003*). For data collected in experiments with *Obp19a*⁻; *Obp28a*⁻ flies and their controls, AC signals were filtered (300–3000 Hz) and amplified with DAM50 (World Precision Instruments), and then the signals were digitized with DIGIDATA 1332A (Axon Instruments) and analyzed

with pClamp 10.3 (Molecular Devices). For the rest of the data, signals were amplified with a 10 × Syntech Universal AC/DC Probe, filtered (100–3000 Hz with 50/60-Hz suppression), and digitized with IDAC-4 (Syntech). Action potentials were detected using AutoSpike 32 software (Syntech). AC signals were recorded for 10 s, starting 1 s before stimulation. Spikes were counted during and before the 500 ms stimulation period, with the 500 ms spike count starting 40 ms after the onset of the stimulus pulse to allow for the transport time of the odor stimulus through the delivery system to the antenna. Responses to stimuli were calculated by subtracting the spike count in the 500 ms of unstimulated activity from the spike count during the 500 ms period of stimulated activity. Responses to diluents alone were not subtracted. Spikes were binned in 25 ms intervals for peristimulus time histograms (PSTHs). Mutant and control flies were tested in parallel in all experiments. n values in figure legends refer to the number of recordings. In general, 1 or 2 sensilla were tested per animal, and a total of 2 or 3 odor pulses were delivered to an animal; in no case were more than 3 sensilla tested or 4 odor pulses delivered.

## Oviposition behavior

Oviposition assays were carried out in Petri dishes containing four quadrants (Dot Scientific, CAT # 557684). Two quadrants contained 0.25% (w/v) agarose with 1% (w/v) sucrose, which was added to elicit oviposition. One of the two quadrants contained a sample of linoleic acid, while the other was a control. To better solubilize the linoleic acid sample, it was first pre-mixed in a 0.2% (v/v) solution of ethanol, which was then added to agarose that was at 55–58°C. After vigorous shaking, 5 ml of this agarose was introduced into a quadrant. The other quadrant contained agarose prepared in exactly the same way, but with 0.2% ethanol that contained no linoleic acid sample.

10 newly eclosed females were cultured with three males in a vial for 5–6 days. Shortly before the assay, flies were immobilized on ice, and female flies were gently placed into the two quadrants that contained no agarose. Ten females were maintained in the Petri dish for 20–24 hr in a dark room (25°C, 60% humidity). An oviposition preference index (OPI) was then calculated as follows: OPI = (number of eggs on stimulus quadrant – number of eggs on control quadrant)/ (total number of eggs). A very small fraction of dishes contained fewer than 10 eggs and were excluded from the assay. Mutant and control flies were tested in parallel in all cases.

## Acknowledgements

We thank Zina Berman for technical support. We thank Dr. Melissa Harrison, Dr. Kate O'Connor-Giles, Dr. Jill Wildonger, and Dr. Simon Bullock for plasmids pDsRed-attP (Addgene plasmid 51019) and pCFD4-U6:1_U6:3tandemgRNAs (Addgene plasmid 49411). We thank Drs. Hany Dweck, Mahmut Demir, Neeraj Soni, and Srinivas Gorur-Shandilya for help with electrophysiology. We thank Dr. Rudi Turner for providing the scanning electron microscope photograph presented in Figure 1A. This work was supported by the China Scholarship Council-Yale World Scholars Program (XS), by an NSF Graduate Research Fellowship (JSS), by the Dwight N and Noyes D Clark Scholarship Fund (JSS), by a Scholar Award from the International Chapter of the PEO Sisterhood (JSS), and by NIH DC02174, NIH DC04729, and NIH DC11697 (JRC)

## Additional information

### Funding

| Funder | Grant reference number | Author |
| --- | --- | --- |
| China Scholarship Council | | Shuke Xiao |
| National Science Foundation | | Jennifer S Sun |
| Dwight N and Noyes D Clark Scholarship Fund | | Jennifer S Sun |
| P.E.O. Scholar Award | | Jennifer S Sun |
| National Institutes of Health | DC02174 | John R Carlson |
| National Institutes of Health | DC04729 | John R Carlson |

| National Institutes of Health | DC11697 | John R Carlson |

The funders had no role in study design, data collection and interpretation, or the decision to submit the work for publication.

## Author contributions

Shuke Xiao, Conceptualization, Resources, Data curation, Formal analysis, Funding acquisition, Investigation, Visualization, Methodology, Writing—original draft, Writing—review and editing; Jennifer S Sun, Resources, Funding acquisition, Investigation, Methodology, Writing—review and editing; John R Carlson, Conceptualization, Resources, Formal analysis, Supervision, Funding acquisition, Methodology, Writing—original draft, Project administration, Writing—review and editing

## Author ORCIDs

Shuke Xiao https://orcid.org/0000-0003-2103-128X
Jennifer S Sun http://orcid.org/0000-0002-4274-0504
John R Carlson https://orcid.org/0000-0002-0244-5180

## Decision letter and Author response

Decision letter https://doi.org/10.7554/eLife.51040.019
Author response https://doi.org/10.7554/eLife.51040.020

# Additional files

## Supplementary files

• Transparent reporting form DOI: https://doi.org/10.7554/eLife.51040.016

## Data availability

All data generated or analysed during this study are included in the manuscript and supporting files. Source data files have been provided for Figure 2, Figure 3, Figure 4, Figure 2—figure supplement 1, Figure 2—figure supplement 2, Figure 3—figure supplement 1.

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
