## [Decision Letter]

Thank you for submitting your article "Robust olfactory responses in the absence of odorant binding proteins" for consideration by *eLife*. Your article has been reviewed by three peer reviewers, and the evaluation has been overseen by Leslie Griffith as Reviewing Editor and Catherine Dulac as the Senior Editor. The reviewers have opted to remain anonymous.

The reviewers have discussed the reviews with one another and the Reviewing Editor has drafted this decision to help you prepare a revised submission.

Summary:

This work builds on a previous publication by this group showing that elimination of OBP28 had little effect on olfactory function in ab8. This was contradictory to the widely held (but little tested) idea that OBPs are critical to the detection of odorants, especially hydrophobic substances. In this paper the authors generate mutants for an additional 3 highly expressed OBPs and demonstrate that there is little functional effect of loss of any or all of these proteins.

Essential revisions:

The reviewers uniformly felt that the paper was of high technical quality and that there was not a need for additional experimentation. There were a number of suggestions that the authors should consider, but the main thing that they need to do to revise this paper is to add some additional commentary on a couple of the issues.

1) The authors should discuss a paper just published in Genetics by Scheuermann and Smith, 2019. Those authors report that Obp83a/b double mutants have no effect on basiconic sensilla (consistent with this report), but instead affect the deactivation kinetics for some odorants in trichoid sensilla. What does this suggest about the role of OBP83a/b? Why do the authors think there is this sensilla-specific difference?

2) What is the likely mechanism for the linoleic acid effect? Do the OBPs bind to linoleic acid? Might this decrease the concentration that reaches the ORN?

3) Authors should further discuss why removing a major protein expressed at such a huge concentration in the sensillum lymph has so little impact on the sensitivity of the receptors, and even in some cases apparently enhances their sensitivity. And perhaps to speculate about the nature of the behavioral differences in some cases.

4) Reviewers wondered if the way olfactory responses are calculated could smooth out differences between normal and mutant flies. In the Materials and methods (subsection “Electrophysiology”), the authors report that the responses were calculated by subtracting the spike count during the 500 ms before stimulation. While this method is relevant when one wants to screen for ligands which are stimulating a given receptor, what happens if you do not do this? The rationale behind this question is that in the absence of OBPs, receptors may become more sensitive to the background of odorants which would not be "removed" as quickly as when OBPs are present. In the same line of thought, if "compensation" has been introduced in the PSTHs of Figures 3C-F and Figure 3—figure supplement 1. Figure 3C and Figure 3—figure supplement 1A, is this suggestive that olfactory responses in mutants flies are adapting less rapidly and show more residual activity than in the wild type flies, at least for pulses 1-3. Is this a consistent pattern for other odorants? Would it be possible to make these differences more visible by subtracting the curves (obp mutant – wild type)?

---

## [Author Response]

Essential revisions:The reviewers uniformly felt that the paper was of high technical quality and that there was not a need for additional experimentation. There were a number of suggestions that the authors should consider, but the main thing that they need to do to revise this paper is to add some additional commentary on a couple of the issues.1) The authors should discuss a paper just published in Genetics by Scheuermann and Smith, 2019. Those authors report that Obp83a/b double mutants have no effect on basiconic sensilla (consistent with this report), but instead affect the deactivation kinetics for some odorants in trichoid sensilla. What does this suggest about the role of OBP83a/b? Why do the authors think there is this sensilla-specific difference?

We have added discussion of this recent paper, which we were not aware of when we submitted our paper. We emphasize that two key conclusions of this paper agree with ours: i) that robust olfactory responses remain after removal of two Obps; ii) that a phenotype is found only in a subset of neurons and with a subset of odorants. We describe the phenotype that Scheuermann and Smith found, and the mechanism they propose for the role of OBP83a/b.

2) What is the likely mechanism for the linoleic acid effect? Do the OBPs bind to linoleic acid? Might this decrease the concentration that reaches the ORN?

We acknowledge that the mechanism is unclear but that, as the reviewer suggests, the phenotype is consistent with a model in which the OBP binds to the odorant and decreases the concentration that reaches the ORN.

3) Authors should further discuss why removing a major protein expressed at such a huge concentration in the sensillum lymph has so little impact on the sensitivity of the receptors, and even in some cases apparently enhances their sensitivity. And perhaps to speculate about the nature of the behavioral differences in some cases.

We agree; the lack of a major impact is perhaps the most surprising result of our study and have added discussion of this point. We suggest the possibility that some of these antennal Obps might play a role in processes other than olfaction, as they do in other organs.

4) Reviewers wondered if the way olfactory responses are calculated could smooth out differences between normal and mutant flies. In the Materials and methods (subsection “Electrophysiology”), the authors report that the responses were calculated by subtracting the spike count during the 500 ms before stimulation. While this method is relevant when one wants to screen for ligands which are stimulating a given receptor, what happens if you do not do this? The rationale behind this question is that in the absence of OBPs, receptors may become more sensitive to the background of odorants which would not be "removed" as quickly as when OBPs are present. In the same line of thought, if "compensation" has been introduced in the PSTHs of Figures 3C-F and Figure 3—figure supplement 1. Figure 3C and Figure 3—figure supplement 1A, is this suggestive that olfactory responses in mutants flies are adapting less rapidly and show more residual activity than in the wild type flies, at least for pulses 1-3. Is this a consistent pattern for other odorants? Would it be possible to make these differences more visible by subtracting the curves (obp mutant – wild type)?

We have examined the spike counts before stimulation, *i.e.* the spontaneous firing levels, and have found that the spontaneous activity levels of all 12 neurons were normal in the absence of abundant Obps (p>0.05, Mann-Whitney U test, n=10). This equivalence of spontaneous firing rates in mutant and control argues against several interesting models. We tried subtracting the curves as suggested. We don't feel confident in drawing conclusions from the replotted data, which are noisy; we prefer to state what we are confident of, *i.e.* that the dynamics of the response throughout the entire stimulus period appeared comparable to that of the control, and to add to the Discussion an acknowledgement that we have not analyzed the response dynamics extensively.